# Fully-exposed Pt-Fe cluster for efficient preferential oxidation of CO towards hydrogen purification

Zhimin Jia[1,2,10], Xuetao Qin[3,10], Yunlei Chen[4,5,10], Xiangbin Cai [6,10], Zirui Gao[3], Mi Peng[3], Fei Huang [1], Dequan Xiao [7], Xiaodong Wen [4,5], Ning Wang [6], Zheng Jiang [8], Wu Zhou [9], Hongyang Liu [1,2] ✉ & Ding Ma [3] ✉

Hydrogen is increasingly being discussed as clean energy for the goal of net-zero carbon emissions, applied in the proton-exchange-membrane fuel cells (PEMFC). The preferential oxidation of CO (PROX) in hydrogen is a promising solution for hydrogen purification to avoid catalysts from being poisoned by the trace amount of CO in hydrogen-rich fuel gas. Here, we report the fabrication of a novel bimetallic Pt-Fe catalyst with ultralow metal loading, in which fully-exposed Pt clusters bonded with neighbor atomically dispersed Fe atoms on the defective graphene surface. The fully-exposed PtFe cluster catalyst could achieve complete elimination of CO through PROX reaction and almost 100% CO selectivity, while maintaining good stability for a long period. It has the mass-specific activity of 6.19 $(mol_{CO})*(g_{Pt})^{-1}*h^{-1}$ at room temperature, which surpasses those reported in literatures. The exhaustive experimental results and theoretical calculations reveal that the construction of fully-exposed bimetallic Pt-Fe cluster catalysts with maximized atomic efficiency and abundant interfacial sites could facilitate oxygen activation on unsaturated Fe species and CO adsorption on electron-rich Pt clusters to hence the probability of CO oxidation, leading to excellent reactivity in practical applications.

Hydrogen, as renewable and clean energy, will play an increasingly important role in meeting the world's growing energy needs and avoiding the potential negative influence to climate change[1]. The proton-exchange membrane fuel cell (PEMFC) has been regarded as a promising candidate for the utilization of hydrogen, due to its high efficiency and low operating temperature[2,3]. However, the platinum electrode of the PEMFC is prone to be poisoned by CO leading to the loss of efficiency, which is attributed to the presence of small amounts of CO remaining in the industrial hydrogen produced by the reforming of hydrocarbons and water-gas shift reaction processes[4,5]. To remove

[1]Shenyang National Laboratory for Materials Science, Institute of Metal Research, Chinese Academy of Sciences, Shenyang 110016, P. R. China. [2]School of Materials Science and Engineering, University of Science and Technology of China, Shenyang 110016, P. R. China. [3]College of Chemistry and Molecular Engineering, Peking University, Beijing 100871, P. R. China. [4]University of Chinese Academy of Science, No. 19A Yuanquan Road, Beijing 100049, P. R. China. [5]State Key Laboratory of Coal Conversion, Institute of Coal Chemistry, Chinese Academy of Sciences, Taiyuan 030001, P. R. China. [6]Department of Physics and Center for Quantum Materials, Hong Kong University of Science and Technology, Clear Water Bay, Kowloon, Hong Kong SAR, P. R. China. [7]Center for Integrative Materials Discovery, Department of Chemistry and Chemical Engineering, University of New Haven, 300 Boston Post Road, West Haven, CT 06516, USA. [8]Shanghai Institute of Applied Physics, Chinese Academy of Sciences, Shanghai 201204, P. R. China. [9]School of Physical Sciences and CAS Center for Excellence in Topological Quantum Computation, University of Chinese Academy of Sciences, Beijing 100049, P. R. China. [10]These authors contributed equally: Zhimin Jia, Xuetao Qin, Yunlei Chen and Xiangbin Cai. ✉e-mail: liuhy@imr.ac.cn; dma@pku.edu.cn

the residual CO before feeding the hydrogen fuel gas to the PEMFC, the CO preferential oxidation (PROX) reaction is considered as an attractive route by selective CO oxidation in $H_2$-rich stream[6,7]. However, since the PEMFC operation temperature is low, it remains challenging to achieve highly efficient oxidation of CO at low temperature.

Among various supported metal catalysts developed for the PROX reaction, platinum group metal (PGM) catalysts with excellent activity and stability have received much attention as potential candidates[8,9]. Unfortunately, it is widely known that monometallic PGM catalysts on inert supports have poor CO oxidation activity in PROX reaction especially at low temperature, e.g., room temperature, due to the limited $O_2$ activation on the Pt surface adsorbed by CO[5,10]. So far, to improve the CO PROX catalytic performance at low temperature, intensive studies have been done over promoted PGM catalysts, in particular, by introducing various secondary reducible element into PGM catalysts[11–13]. One of the promoted catalysts is bimetallic alloy nanoparticles, in which the second element and Pt could form an intermetallic compound with their own specific crystal structure[13,14]. Chen and co-workers prepared a PtFe nano-alloy catalyst by a one-pot surfactant-free polyol process. They discovered that the high performance of $Pt_{0.71}Fe_{0.23}/Al_2O_3$ towards CO PROX reaction originated from the adsorption of CO and $O_2$ on Pt and Fe, respectively, leading to the efficient $O_2$ activation on $Fe^{2+}$ species. But these active species would be oxidized to $Fe^{3+}$ under the PROX condition, causing the deactivation of supported PtFe alloy catalysts[15]. Another type of promoted catalysts is reducible metal oxides-promoted PGM catalysts that use reducible oxides as supports, such as $CeO_2$, $Fe_2O_3$, and $Co_3O_4$, sometimes with the presence of secondary metal (Fe, Co, Ni, Cu) as promoter[16–18]. Zhang et al. reported a novel catalyst consisting of only single Pt atoms that were uniformly dispersed on $FeO_x$ nanocrystallites with high activity for preferential oxidation of CO in $H_2$ due to the surface oxygen vacancy of iron oxides and the strong binding of Pt atoms with positive charge[12]. Recently, Lu et al. showed that atomically dispersed iron hydroxides on Pt nanoparticles using the atomic layer deposition (ALD) method enabled completely selective CO removal through the PROX reaction. The $Fe_1(OH)_x$-Pt single interfacial sites could readily react with CO and facilitate oxygen activation[19]. Overall, the promoted PGM catalysts by the addition of a secondary element have exhibited improved oxidation activities at low temperature for the PROX reaction due to the presence of additional accessible sites for oxygen such as oxygen vacancy or reducible metal oxides/hydroxides provided by secondary metal sites to further react with CO. Nonetheless, the main species in these promoted catalysts mentioned above were commonly the metal nanoparticles over 2 nm in size which did not achieve the optimal utilization of the metal atoms in PtM catalysts. The prepared promoted PGM catalysts only offered a limited number of interfacial sites for the adsorption of CO and $O_2$, and it is usually hard to maximize the interfacial density. Thus, the insights are in urgent need into achieving the size of metal species with sub-nano level or even atomic level. Besides, only a few catalysts reported in literatures showed superior oxidation activity and selectivity at low temperature, particularly, at room temperature, which is critical for PEMFC operation conditions[6,20]. In recent years, fully exposed cluster catalysts (FECCs) with ensemble sites and atomic dispersion have drew attention as active species in many catalytic reactions[21–23]. As a result, it is of great interests to construct highly dispersed and fully exploited interfacial sites via the addition of the secondary element in hopes of improving the catalytic performance for CO PROX reaction at low temperature.

In this work, we reported a novel catalyst, fully-exposed bimetallic Pt-Fe clusters anchored on the defective graphene/nanodiamond (ND@G) hybrid support, where the atomically dispersed and fully-exposed Pt cluster are bonded with neighboring Fe atom to provide abundant Pt-Fe interfaces. The fully-exposed Pt-Fe clusters are highly efficient for preferential oxidation of CO with complete CO conversion and 100% CO selectivity at low temperature, and show excellent oxidation activity (high CO oxidation mass-specific reaction rate of 6.19 $(mol_{CO})*(g_{Pt})^{-1}*h^{-1}$ at 30 °C) with ultralow metal loading. The fully-exposed Pt-Fe clusters with high atomic efficiency of both two metals and abundant Pt-Fe active interfaces contribute to the improved catalytic performance. Besides, the distinctive Pt-Fe interfacial sites can activate CO and $O_2$ molecules to accelerate CO oxidation with lowered apparent activation energy.

## Results

### Structural characterizations of PtFe cluster catalysts

Fully-exposed Pt-Fe clusters were fabricated on defective graphene/nanodiamond (ND@G) hybrid support through a facile deposition-precipitation method with theoretical loadings (Pt: 0.75 wt%, Fe: 0.2 wt %), which was denoted as 0.75Pt0.2Fe/ND@G (Table S1). The morphology and structure of 0.75Pt0.2Fe/ND@G were initially investigated by Aberration-corrected high-angle annular dark-field scanning transmission electron microscope (HAADF-STEM). As revealed in Figs. 1a, 1b and Fig. S1, ultrasmall Pt-Fe clusters were fully exposed and uniformly dispersed on the ND@G support, without any crystallized platinum or iron nanoparticles[21,24]. Meanwhile, the X-ray diffraction (XRD) pattern of the catalyst (Fig. S2) showed only typical diffraction peaks of graphene and nanodiamond[25,26], confirming that no crystalline metal was formed, in line with the STEM images of 0.75Pt0.2Fe/ ND@G. The corresponding energy dispersive X-ray (EDX) mapping results of 0.75Pt0.2Fe/ND@G (Fig. 1c) indicated that platinum and iron element displayed simultaneously within the cluster areas on the support surface. The electron energy loss spectroscopy (EELS) (Fig. S3) further provided strong evidence of the presence of atomically dispersed iron atom structure in the cluster area, indicating that the Pt-Fe clusters mainly consisted of fully-exposed Pt clusters and highly dispersed Fe atoms. As shown in the magnified HAADF-STEM images (Fig. 1d), several compact spots, with different brightness caused by different Z-contrast of Pt and Fe, were packed loosely together to form atomic clusters marked by the rectangles, which consisted of several (mostly 3-5) Pt atoms and a neighboring Fe atom. Such clusters were distributed homogeneously on the surface of the ND@G substrate. The intensity surface plot images of circled clusters in Fig. 1e, f intuitively distinguished the Pt and Fe atoms due to the difference of signal intensity contribution between Pt and Fe atoms, confirming the formation of fully-exposed Pt-Fe clusters. Using a similar synthetic method, the 0.75Pt/ND@G catalyst with monometallic platinum clusters was obtained. From STEM images and intensity profiles analysis of 0.75Pt/ND@G (Fig. S4), the similar cluster structure with single-atom-layer thickness highly and disorderly dispersed on the support[27,28].

To probe the electronic structure of the fully-exposed Pt-Fe cluster for 0.75Pt0.2Fe/ND@G, X-ray photoelectron spectroscopy (XPS) and X-ray absorption spectrometric (XAS) measurements were performed. The XPS analysis (Fig. S5) showed that the Pt $4f$ peak of 0.75Pt0.2Fe/ND@G was shifted to lower binding energy compared to that of 0.75Pt/ND@G, indicating that Pt electronic states were modified by Fe in fully-exposed Pt-Fe clusters via electron transfer from Fe to Pt atoms[14,29]. Temperature programming desorption of CO (CO-TPD) experiments (Fig. S6) further suggested that 0.75Pt0.2Fe/ND@G catalyst had higher CO desorption temperature than 0.75Pt/ND@G due to electron-rich Pt clusters, in agreement with the XPS results. From the normalized X-ray absorption near-edge structure (XANES) curves at the Pt-$L_3$ edge (Fig. 2a), the white line intensity of 0.75Pt0.2Fe/ND@G decreased in compared to that of 0.75Pt/ND@G, which could be attributed to the electron-rich Pt clusters in 0.75Pt0.2Fe/ND@G[30]. Correspondingly, the 0.75Pt0.2Fe/ND@G catalyst exhibited the higher absorption energy than 0.2Fe/ND@G at the Fe $K$-edge (Fig. 2b), indicating the electron-deficient feature of Fe atoms in 0.75Pt0.2Fe/ND@G. Furthermore, Fourier-transform extended X-ray absorption fine structure (FT-EXAFS) spectra (Figs. 2c and

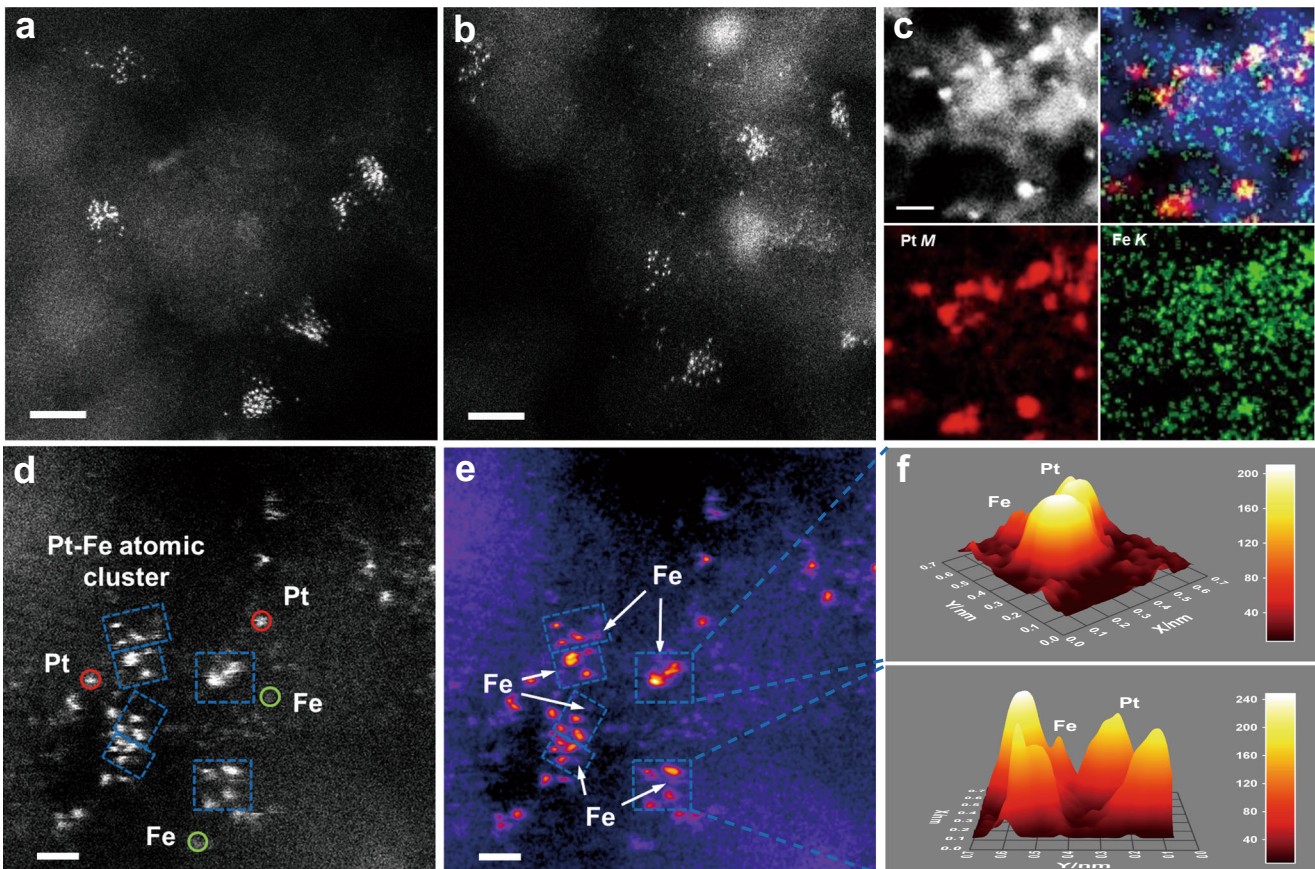

**Fig. 1 | Microscopic structures of 0.75Pt0.2Fe/ND@G. a, b** HAADF-STEM images of 0.75Pt0.2Fe/ND@G at high magnification. **c** Energy-dispersive X-ray (EDX) mapping images of 0.75Pt0.2Fe/ND@G catalyst. **d, e** The high resolution HAADF-STEM image of the 0.75Pt0.2Fe/ND@G, where representative isolated Pt, Fe atoms and atomic Pt-Fe clusters are highlighted by red, green circles and dashed blue rectangles, respectively. **f** 3-D intensity surface plot and intensity range shown for the dashed regions of the image in **e**. Scale bar: **a**, 2 nm; **b** 2 nm; **c**, 2 nm; **d** 0.5 nm; **e** 0.5 nm.

2d) were provided to clarify the local configurations of atomically dispersed Pt and Fe in fully-exposed Pt-Fe clusters. 0.75Pt0.2Fe/ND@G and 0.75Pt/ND@G had the same weak peaks at 2.00 Å and 2.70 Å at Pt $L_3$-edge attributed to Pt-O/C and Pt-Pt paths, respectively, indicating high dispersion of Pt species. Besides, the first Fe-O/C peak located at around 1.96 Å displayed both 0.75Pt0.2Fe/ND@G and 0.2Fe/ND@G with the same Fe loading prepared by the similar method at Fe $K$-edge. It was evidenced that Pt and Fe species could be anchored on ND@G through M(Pt/Fe)-C bonding, which was further verified by the appearance of M(Pt/Fe)-C peak at 282.9 eV in C 1$s$ XPS spectra (Fig. S7)[31]. Interestingly, the second coordination peak at 2.56 Å of 0.75Pt0.2Fe/ND@G showed a shift compared to that of 0.2Fe/ND@G assigned to the Fe-Fe path according to Fe foil. It inferred the change of neighbor metal atom shell around the Fe atom, demonstrating the formation of Pt-Fe coordination. Then, the obtained EXAFS fitting results for 0.75Pt0.2Fe/ND@G (Fig. 2e-f, Fig. S8 and Table S2–3) showed that the coordination number of Pt-Fe and Pt-Pt were 0.6 and 2.5 at Pt $L_3$-edge, and the coordination number of Pt-Fe and Fe-Fe were 3.4 and 1.3 at Fe $K$-edge, respectively[32]. Combined with the STEM images, the atomically dispersed and fully-exposed Pt clusters bonded with iron species that were primarily present in the atomically dispersed state. Such atomic configurations featured with the fully-exposed Pt-Fe clusters via Pt-Fe interfaces that were immobilized on the support by M(Pt/Fe)-C bonds. To corroborate this result, the wavelet transform (WT) analysis of Fe EXAFS was carried out[33–35]. As shown in Fig. 2g, 0.2Fe/ND@G displayed the intensity maximum at 5.0 Å$^{-1}$ and 7.9 Å$^{-1}$, which were attributed to the Fe-O/C and Fe-Fe paths, respectively, according to the WT contour plots of Fe foil and Fe$_2$O$_3$

references. In contrast, for the WT signal of 0.75Pt0.2Fe/ND@G, one intensity maximum at around 6.1 Å$^{-1}$ was exclusively observed, which was assigned to the Pt-Fe coordination.

## Catalytic performance for PROX oxidation
The preferential oxidation of CO in hydrogen (PROX) was studied to understand the role of fully-exposed Pt-Fe clusters. The 0.75Pt/ND@G catalyst with similar platinum structures of atomic clusters could reach a maximum CO conversion of 85% at 60 °C (Fig. S11). In contrast, by introducing Fe species, the 0.75Pt0.2Fe/ND@G catalyst dramatically improved the oxidation activity by achieving a full CO conversion at around room temperature (30 °C), with a high space velocity of 45,000 mL g$^{-1}$ h$^{-1}$ and 100% CO selectivity (Fig. 3a). However, 0.75Pt0.2Fe/SiO$_2$ with the same Pt and Fe loadings showed low activity at room temperature, due to the presence of unevenly distributed metal nanoparticles which were observed in the STEM and XRD results of 0.75Pt0.2Fe/SiO$_2$ (Fig. S9–10). In addition, the mass-specific reaction rate of 0.75Pt0.2Fe/ND@G at 30 °C was 6.19 (mol$_{CO}$)*(g$_{Pt}$)$^{-1}$*h$^{-1}$, which was higher than those of noble-metal catalysts reported in literatures (Fig. 3b, c and Table S4). The good stability of the PROX reaction at low CO conversion could be evaluated for more than 100 h at 30 °C for the 0.75Pt0.2Fe/ND@G catalyst (Fig. 3d). In the presence of water and CO$_2$ (Fig. S12), the catalyst was still stable in the reaction stream without obvious activity drop. After the PROX reaction, we employed a comprehensive structure characterization of the used catalyst. Combined with the STEM images and XAS results (Fig. S13–14), the detailed structure of used 0.75Pt0.2Fe/ND@G catalyst was still better maintained with typical fully-exposed Pt-Fe clusters. The iron species were

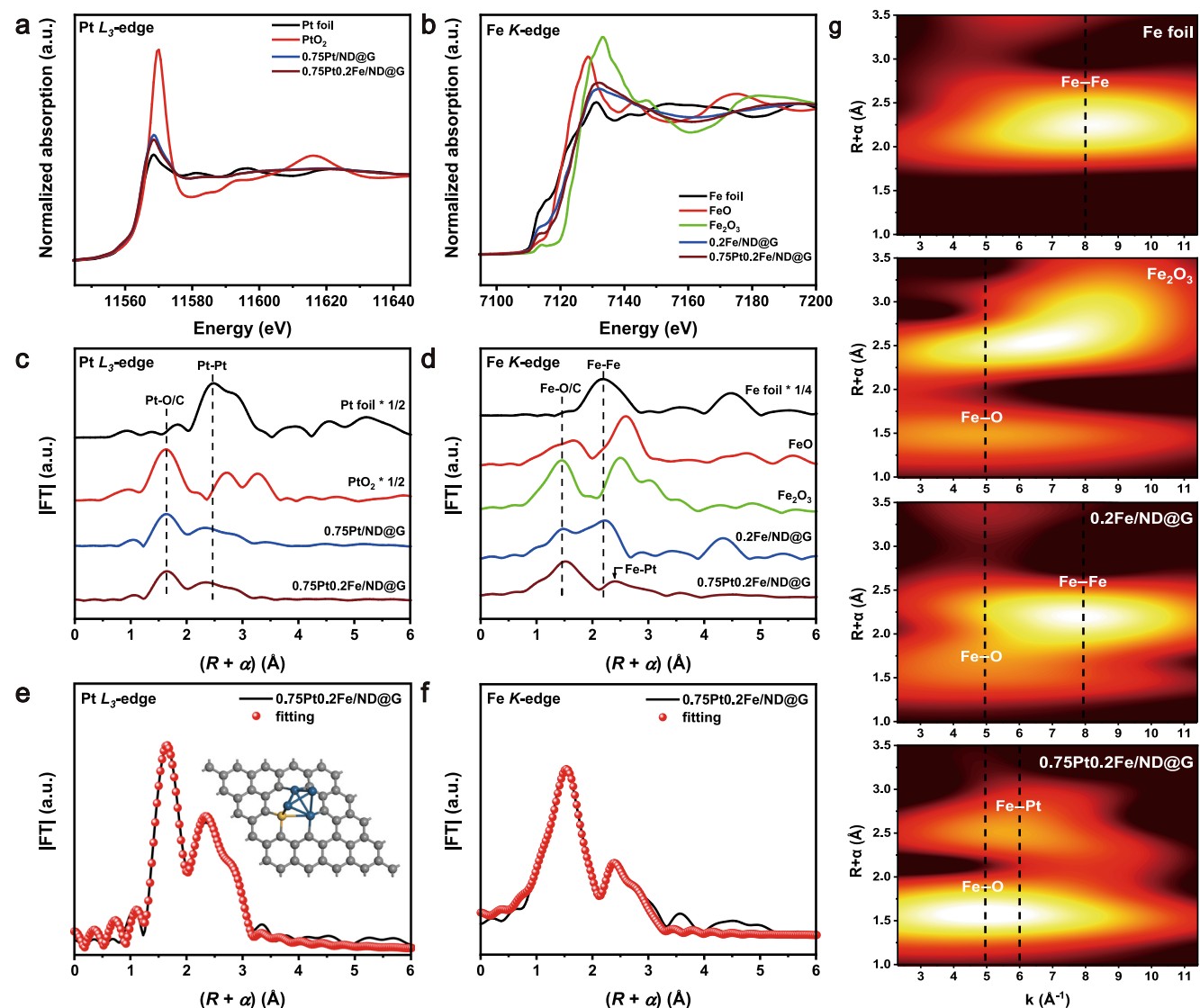

**Fig. 2 | Synchrotron XAFS measurements of catalysts. a** XANES and **c** FT-EXAFS spectra at the Pt $L_3$-edge (without phase correction) of the 0.75Pt/ND@G and 0.75Pt0.2Fe/ND@G; **b** XANES spectra and **d** FT-EXAFS at the Fe $K$-edge of the 0.2Fe/ ND@G and 0.75Pt0.2Fe/ND@G. **e, f** The corresponding Pt $L_3$-edge and Fe $K$-edge EXAFS fitting curves for 0.75Pt0.2Fe/ND@G catalyst at R-space, respectively. **g** Fe $K$-edge WT-EXAFS of 0.2Fe/ND@G, 0.75Pt0.2Fe/ND@G and the references samples.

slightly oxidized after the PROX reaction, as observed in the Fe $K$-edge XAS spectra for the used catalyst. Notably, the 0.75Pt0.2Fe/ND@G catalyst could also achieve 100% CO conversion at low temperature under the PROX reaction with excess oxygen (Fig. S15). Finally, as a side note, the 0.75Pt0.2Fe/ND@G catalyst exhibited excellent performance for CO oxidation in the absence of hydrogen with a high TOF at 30 °C in comparison with other reported catalysts and robust stability (Fig. S16 and Table S5).

## CO oxidation mechanism studies

To verify the promoted oxidation activity caused by the Pt-Fe interfacial effect, we evaluated the catalytic performance in the CO PROX reaction by regulating the Fe loading from 0.1 wt% to 0.3 wt%. Compared with 0.75Pt/ND@G, 0.75Pt0.1Fe/ND@G showed an increasing activity at 30 °C due to the introduction of a small amount of iron, and the oxidation activity reached maximum with the Fe dopant of 0.2 wt%, whereas a further increase of Fe loading decreased the performance (Fig. 4a, Fig. S17 and Table S1). The improved reaction rate after the introduction of iron for these PtFe catalysts suggested that the Pt-Fe interfacial sites were speculated as the active centers. Kinetic studies of

the catalysts with different Fe loading revealed that the similar apparent activation barriers for a series of PtFe catalysts supported on ND@G distinctly were lower than that of 0.75Pt/ND@G, which might indicate the different reaction mechanism or active site among those PtFe catalysts (Fig. 4b). The structures of catalysts with different Fe loadings were further investigated (Fig. S18–24, and Table S2–3). The XRD results revealed no diffraction peak of metals appeared other than those corresponding to the support, indicating that Pt-Fe species were highly dispersed on ND@G for PtFe catalysts with different Fe loadings. Specifically, for the 0.75Pt0.1Fe/ND@G, a similar structure was elaborated from the STEM images. The related EDX mapping images (Fig. S19–20) and the XAS results (Fig. S23) were similar with those of 0.75Pt0.2Fe/ND@G, suggesting that the fully-exposed Pt-Fe clusters with Pt-Fe interfacial sites could react efficiently with CO and $O_2$. However, for the 0.75Pt0.3Fe/ND@G, more iron species around clusters and ND@G-supported iron species were observed in the STEM images and EDX mapping results (Fig. S21–22), except for the fully-exposed Pt-Fe clusters. Meanwhile, the EXAFS results showed the Fe-O-Fe coordination peak at the high shell was increased, suggesting the formation of iron clusters in line with the STEM of 0.75Pt0.3Fe/ND@G.

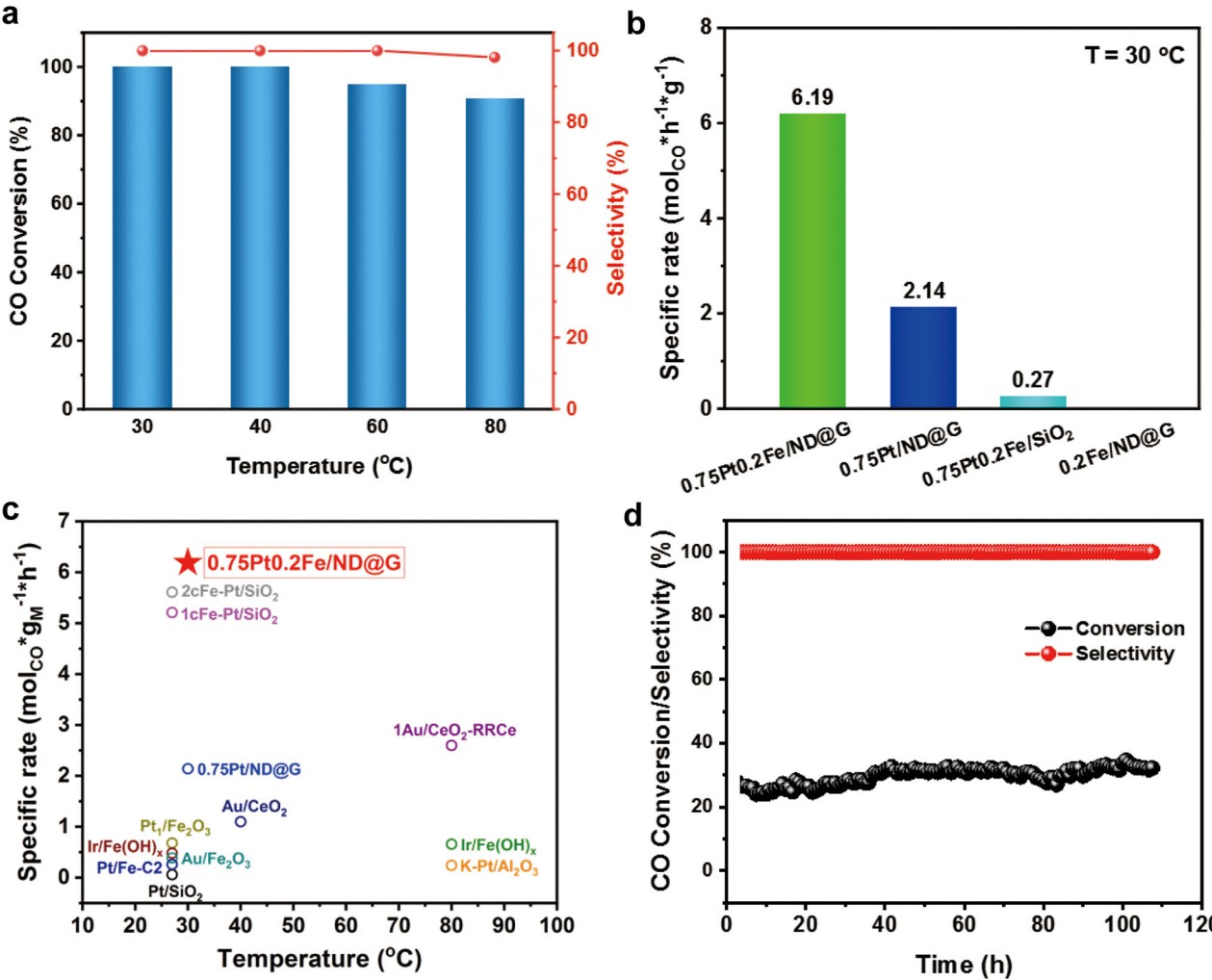

**Fig. 3 | Catalytic performance for the PROX reaction. a** CO conversion and selectivity of 0.75Pt0.2Fe/ND@G catalyst in PROX reaction. Reaction conditions: 1% CO, 0.5% $O_2$ and 48% $H_2$ balanced in He; the space velocity is 45000 mL $g^{-1}$ $h^{-1}$. **b** Mass specific rates of various PtFe catalysts at 30 °C in PROX reaction. **c** Comparison of the specific rate with reported metal catalysts as a function of their reaction temperature in PROX reaction. **d** Stability of the 0.75Pt0.2Fe/ND@G catalyst in the PROX reaction under the high space velocity (GHSV = 180,000 mL $g^{-1}$ $h^{-1}$) at 30 °C.

Fe $2p$ XPS patterns (Fig. S24) showed more oxidized Fe species for 0.75Pt0.3Fe/ND@G. Therefore, it was presumed that the weakened performance in 0.75Pt0.3Fe/ND@G catalyst could be attributed to the aggregation of many iron species and the decreased number of isolated Pt-Fe interfacial sites.

Moreover, to explore the promotional effects of iron species for PtFe/ND@G catalysts, we conducted the in situ DRIFTS experiments to study the adsorption of reacting molecules[15,36]. For 0.75Pt/ND@G, the main CO chemisorption peak located at 2061 $cm^{-1}$ in the broad peak, assigned to linear CO on Pt clusters (Fig. S25a). However, the addition of iron led to the red shift of linear adsorbed CO signal to 2051 $cm^{-1}$ for 0.75Pt0.2Fe/ND@G (Fig. 4c), confirming the charge transfer from Fe to Pt in line with the XPS and XANE results. We also employed in situ DRIFTS experiments under the reaction condition (1% CO, 1% $O_2$/He flow). The blue shift of linear CO occurred over 0.75Pt/ND@G after the introduction of oxygen (2061 $cm^{-1}$ to 2067 $cm^{-1}$), which reflected the decreased back-donation of electrons from Pt to CO (Fig. S25). This result indicated the co-adsorption of CO and $O_2$ on Pt clusters[16]. As shown in Fig. 4d, the CO adsorption band was slight shift (2054 $cm^{-1}$) in the presence of oxygen compared to the band (2051 $cm^{-1}$) after CO adsorption, which suggested that $O_2$ was adsorbed more likely on the Pt-Fe interface and had a weak effect on Pt sites. The in situ XPS results were collected after hydrogen

reduction at 400 °C and 30 min of time-on-stream in CO oxidation (30 °C) to study the real metal species states during the reaction. The shift to the higher binding energy of Pt $4f$ (Fig. 4e) indicated that Pt species could be slightly oxidized on Pt-Fe interfacial sites after PROX reaction, which was consistent with the in situ DRIFTS results. The Fe $2p_{3/2}$ peaks of 0.75Pt0.2Fe/ND@G catalyst after reduction (Fig. 4f) mainly were located at 706.9 eV and 711.5 eV, ascribed to $Fe_xC$ and $Fe^{3+}$.[37,38] These peaks could be explained by the existence of iron species with positive charge in Fe XAS and metal-C bonding in C XPS results, indicating that iron species of fully-exposed Pt-Fe clusters were anchored by the surrounding carbon on the ND@G surface and other iron oxide species were difficult to reduce on the support. After the PROX reaction for 30 min, a new Fe $2p_{3/2}$ peak at 709.8 eV appeared, indicating that Fe species of fully-exposed Pt-Fe clusters were presented as $Fe^{2+}$ for the activation of oxygen as active sites during the PROX reaction, as reported previously.[15,17] Therefore, the activated oxygen species on Fe atoms readily reacted with CO adsorbed on Pt clusters for interfacial Pt-Fe active sites of fully-exposed Pt-Fe clusters, leading to the outstanding catalytic performance[15,29]. Combined with in situ DRIFTS and XPS results, the obtained relationship between Pt-Fe structures and their properties for the CO PROX reaction with different iron content verified that identified Pt-Fe interfacial sites in fully-exposed bimetallic Pt-Fe

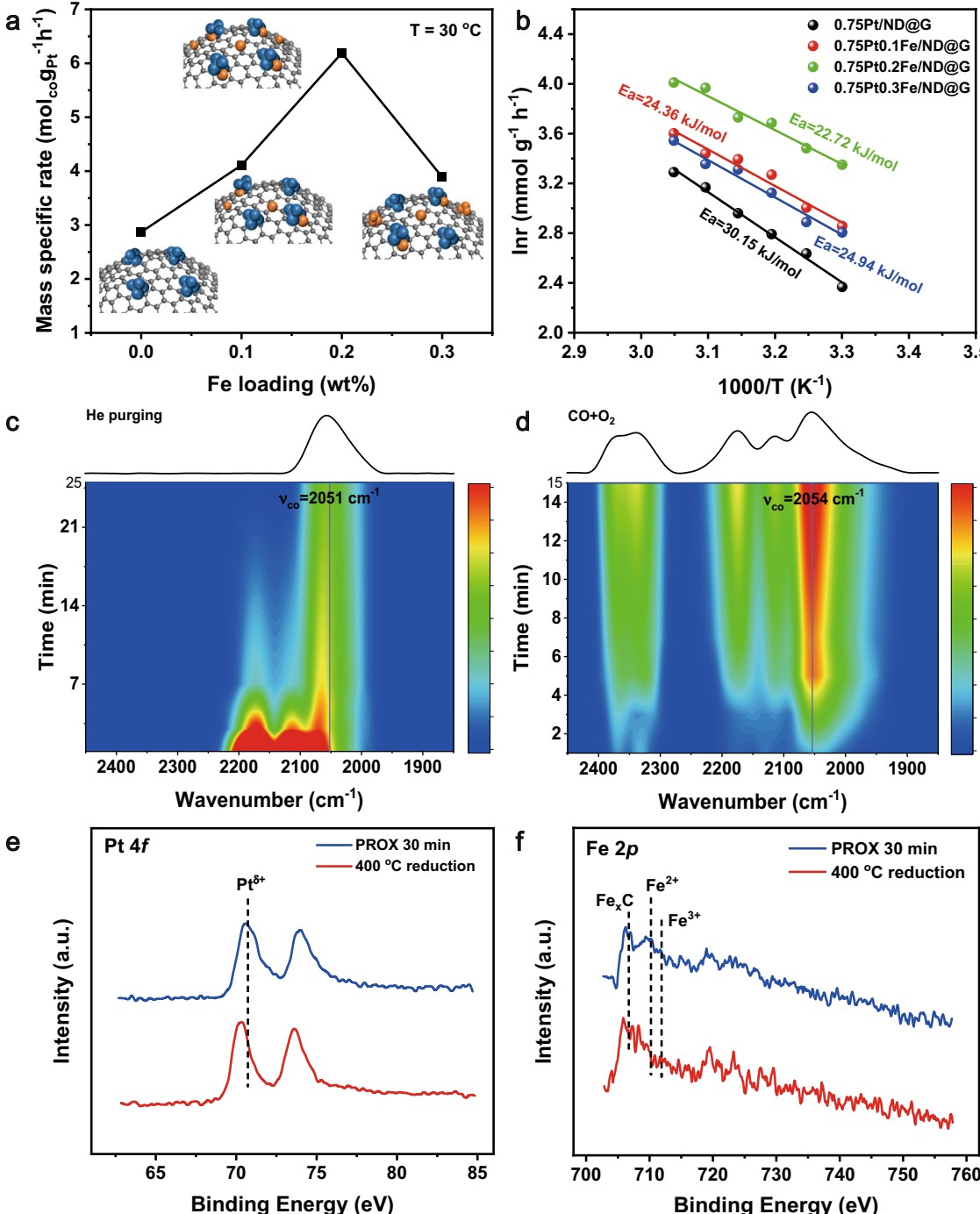

**Fig. 4 | Catalytic mechanism of the 0.75Pt0.2Fe catalyst. a** The correlation between Fe loading of PtFe/ND@G catalysts and catalytic performance for PROX reaction at 30 °C. **b** Apparent activation energies of various PtFe/ND@G catalysts with different Fe loading in PROX reaction. **c** In situ DRIFTS study of CO adsorption during He purging and **d** co-adsorption CO and O₂ on 0.75Pt0.2Fe/ND@G catalyst. XPS spectra of 0.75Pt0.2Fe/ND@G catalyst after in situ reduction at 400 °C and followed by the introduction of reactant gases (1% CO, 0.5% O₂ and 48% H₂ balanced with He at 30 °C for Pt 4$f$ (**e**) and Fe 2$p$ (**f**)).

clusters were highly efficient for the CO PROX reaction at low temperature.

## DFT calculations
To elucidate the origin of promoted oxidation activity of the 0.75Pt0.2Fe/ND@G catalyst, density functional theory (DFT) calculations were carried out to understand the CO oxidation mechanism of fully-exposed Pt-Fe clusters. Firstly, based on the structural characterization, a Pt₄Fe₁@Gr model with four platinum atoms and an iron atom anchored on four-carbon-vacancy of graphene was

constructed to simulate the fully-exposed Pt-Fe cluster on ND@G. Three alternative Pt₄Fe₁ candidate structures were also examined which were later ruled out due to the conflicts between the experimental results of coordination numbers and calculated energy (Fig. 5a,b and Table S6). To further unravel the local structure of Pt₄Fe₁@Gr, we investigated the thermodynamic stability of Pt₄Fe₁@Gr involving oxygen species. Considering the experimental condition with hydrogen reduction treatment at high temperature, it was observed that oxygen species were not stable on Pt₄Fe₁@Gr to form H₂O (Table S7 and S8).

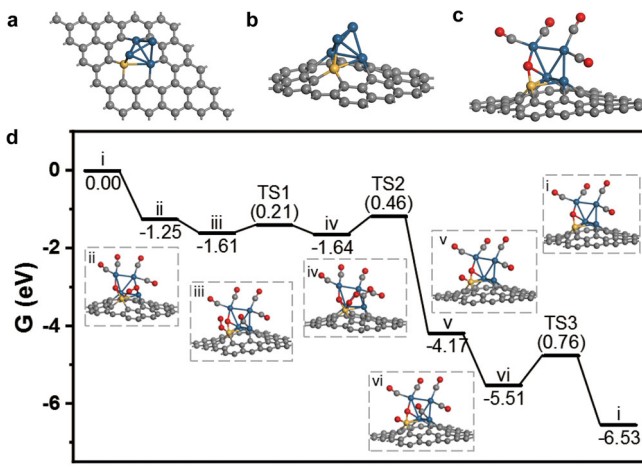

**Fig. 5 | DFT calculations of CO oxidation on 0.75Pt0.2Fe/ND@G. a** Top view and **b** side view of the DFT-optimized structures of 0.75Pt0.2Fe/ND@G after reduction. **c** DFT optimized structure of $Pt_4Fe_1O@Gr$ model. **d** Energy profile of the CO oxidation on a $Pt_4Fe_1@Gr$ model and the structures of intermediates and transition states. Here the Fe, O, C and Pt atoms are in yellow, red, gray and blue, respectively.

On the $Pt_4Fe_1@Gr$ model catalyst, CO preferred to adsorb on Pt ensemble sites while $O_2$ was adsorbed on the Pt-Fe interface (Fig. S26 and Table S9). Interestingly, when CO molecules were adsorbed to a high converge on Pt sites (5CO*), the adsorbed oxygen was easily dissociated with a low barrier of 0.1 eV, of which one O* was embedded in $Pt_4Fe_1$ to form a stable active structure ($Pt_4Fe_1O@Gr$) and other O* reacted readily with CO adsorbed on adjacent Pt site to release $CO_2$ with a barrier of 0.56 eV (Fig. 5c, Fig. S27 and Table S10). Subsequently, on the carbon vacancy anchored Pt-Fe interface, adsorbed CO and $O_2$ readily reacted to generate OCOO* intermediate with a barrier of 0.21 eV and then leaded to the formation of $CO_2$ with a barrier of 0.46 eV (Fig. 5d, Fig. S28 and Table S11). This reaction was exothermic by 2.53 eV. Finally, another CO was adsorbed on the Pt site and attacked the remaining O atom on the Fe site to produce the second $CO_2$ with a barrier of 0.76 eV and a reaction energy of −1.02 eV. Simultaneously, the catalytic reaction cycle was completed and the calculation indicated that this step was rate-determining.

## Discussion

We firstly synthesized fully-exposed Pt-Fe clusters anchored on the defective graphene surfaces via a facile deposition-precipitation method. The structure analysis suggested Pt-Fe interfacial sites were fabricated as active sites by several platinum atoms bonded with neighboring iron atom in the bimetallic Pt-Fe cluster. The fully-exposed Pt-Fe cluster catalyst exhibited excellent catalytic performance for the PROX reaction with 100% CO conversion and selectivity at low temperature, especially with the better mass-specific activity than those reported so far at room temperature, while maintaining good stability for a long period. A fully-exposed Pt-Fe cluster with maximized atomic utilization and abundant interfacial sites could facilitate $O_2$ activated by the neighboring unsaturated Fe site and readily react with adsorbed CO on the fully-exposed Pt cluster to form carbon dioxide on the Pt-Fe interfacial sites, thus resulting in the excellent catalytic performance.

## Methods
### Catalysts preparation
ND@G was prepared by the calcination of nanodiamond (ND) at 1100 °C for 4 h in argon to reconstruct to the defect-rich graphene surface. The PtFe/ND@G catalysts with different Fe loadings (wt(Fe) = 0.1%, 0.2%, 0.3%) were synthesized by a deposition-precipitation method. Firstly, 200 mg ND@G was dispersed in 25 ml deionized water to achieve a homogeneous suspension under 30 min sonication. Then, the ND@G suspension was heated to 100 °C in oil bath followed by the addition of 0.65 g sodium formate. A solution of $H_2PtCl_6 \cdot 6H_2O$ and $Fe(NO_3)_3 \cdot 9H_2O$ was added dropwise into the suspension under continuous stirring for 1 h. Afterwards, the mixture was cooled on standing for 6 h, collected by filter and dried at 60 °C for 12 h under vacuum. Finally, the powders were reduced in hydrogen flow (10% $H_2$/He) at 400 °C for 1 h. The Pt and Fe contents controlled on demand during the preparation process were 0.75 wt% Pt and 0.1, 0.2, 0.3 wt% Fe, respectively, corresponding to 0.75Pt0.1Fe/ND@G, 0.75Pt0.2Fe/ND@G, and 0.75Pt0.3Fe/ND@G. Similarly, the 0.75Pt/ND@G with 0.75 wt% Pt and 0.2Fe/ND@G with 0.2 wt% Fe catalysts were prepared under identical synthesis conditions, but without $Fe(NO_3)_3 \cdot 9H_2O$ and $H_2PtCl_6 \cdot 6H_2O$, respectively. As comparison, we also prepared 0.75Pt0.2Fe/$SiO_2$ (0.75 wt% Pt and 0.2 wt% Fe) using the impregnation method.

### Characterizations
HAADF-STEM images were recorded by a JEOL JEM ARM 200CF Cs-corrected cold field-emission scanning transmission electron microscope at 200 kV accelerating voltage. The electron energy loss spectra (EELS) was carried out using Nion HERMES100 electron microscopy operated at 60 kV. X-ray diffraction (XRD) patterns were collected by an X-ray diffractometer (Bruker Smart APEX II) using a Cu Kα source at a scan rate of 2° $min^{-1}$. The X-ray Photoelectron Spectroscopy (XPS) was performed at ESCALAB 250 instrument with Al Kα radiation. All spectra were calibrated based on the $sp^3$ carbon feature (286.8 eV). The dispersion of Pt species on catalysts was measured by the $H_2$-$O_2$ titration on a Micromeritics AutoChem II 2920 apparatus equipped with a thermal conductive detector (TCD). The actual platinum and iron contents of the catalysts were determined by inductively coupled plasma atomic emission spectrometer (ICP-AES, Leeman-Prodigy 7). The X-ray absorption fine structures (XAFS) measurements were analyzed at the BL14W1 station in Shanghai Synchrotron Radiation Facility (SSRF, 3.5 GeV, 250 mA in maximum, Si (111) double-crystals). The energy was calibrated accordingly to the absorption edge of pure Pt and Fe foil. Before XAFS experiments, the as-prepared samples were transferred to a glove box for tableting without exposure to air. In situ diffuse reflectance infrared Fourier transform spectroscopy (DRIFTS) experiments of 0.75Pt/ND@G and 0.75Pt0.2Fe/ND@G were performed on a Thermo Scientific Nicolet IS10 Fourier transform infrared spectrometer equipped with an MCT detector. The samples were pretreated in 10%$H_2$/He at 400 °C for 1 h and cooled down to 30 °C under He flow. The corresponding spectrums were recorded for 25 min during He purging after CO saturated adsorption for 30 min. Then the catalysts were exposed to the reaction gas (1%CO, 1%$O_2$/He) for 30 min and the spectrums were connected. All results were obtained by averaging 64 scans with a resolution of 4 $cm^{-1}$.

### Reaction evaluation
The PROX reactions were carried out in a quartz-bed flow reactor, equipped with an online gas chromatography instrument (Agilent 7890B with a TCD detector). 20 mg catalysts diluted with 200 mg quartz sands were reduced in hydrogen flow (10% $H_2$/He) at 400 °C for 1 h. The PROX reaction was conducted in a temperature range of 30 °C −200 °C, where the reaction gas feed consisted of 1%CO, 0.5%/1% $O_2$ and 48% $H_2$ in helium at a total flow rate of 15 mL/min. CO oxidation was carried out with 1 vol% CO and 1 vol% $O_2$ with a balance of He, and the space velocity is 72,000 mL $g^{-1}$ $h^{-1}$. Kinetic measurements were performed in the same reactor. The mass specific rates and activation energy were acquired with the CO conversion below 20% by increasing the space velocity.

## Computational details

All DFT calculations were performed with the VASP package[39,40]. The electron−ion interactions were described with the projected augmented waves (PAW) method[41,42]. The PBE exchange-correlation was used[43,44]. The gamma point was used in the Brillouin zone sampling[45]. The plane-wave cutoff energy used was 400 eV. The structural optimization of all the atoms was carried out using a convergence criterion of 0.03 eV/Å for the atomic forces. Gaussian smearing with 0.05 eV was applied for the Brillouin-zone integration. Transition state searches were performed using the automated relaxed potential energy surface scan[46] and harmonic frequency calculations were employed to ensure that the optimized transition states possess exactly one imaginary frequency.

## Data availability

The data supporting this article and other findings are available from the corresponding authors upon request. Source data are provided with this paper.

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

## Acknowledgements

This work was supported by the National Key R&D Program of China (2021YFA1502802, 2021YFA1501100), the National Natural Science Foundation of China (21961160722, 92145301, U21B2092, 91845201, 22072162, 21725301 and 21932002), the Liaoning Revitalization Talents Program XLYC1907055, the Dalian National Lab for Clean Energy (DNL Cooperation Fund 202001) and China Petroleum & Chemical Corporation (No. 420043-2). F. H. acknowledges the support from China Postdoctoral Science Foundation (2021M703279) and the Innovation Foundation from Institute of Metal Research (E255L902A1). N. W. hereby acknowledges the funding support from the Research Grants Council of Hong Kong (Project Nos. C6021-14E, N_HKUST624/19 and 16306818). D.M. acknowledges support from the Tencent Foundation through the XPLORER PRIZE. The XAS experiments were conducted in Shanghai Synchrotron Radiation Facility (SSRF) and Beijing Synchrotron Radiation Facility (BSRF).

## Author contributions

H.L. and D.M. conceived the project. Zhi.J. conducted material synthesis and performed the reaction tests. Y.C. and X.W. did the DFT calculations. X.C. and N.W. contributed to the electron microscopy (HAADF-STEM) study. Z.G. and W.Z. carried out the electron energy loss spectroscopy analysis. M.P. and J.X. conducted the X-ray photoelectron spectroscopy. X.Q. and Zhe.J. conducted the X-ray absorption fine structure spectroscopic measurements and analyzed the data. The manuscript was primarily written by Zhi.J., Y.C., D.X., H.L. and D.M. All authors contributed to discussions and manuscript review.

## Competing interests

The authors declare no competing interests.
