## [Peer Review File · Nature Communications]

Fully-exposed Pt-Fe cluster for efficient preferential oxidation of CO towards hydrogen purificationREVIEWER COMMENTS

Reviewer #1 (Remarks to the Author):

The manuscript “Fully-exposed Pt-Fe cluster for efficient preferential oxidation of CO towards hydrogen purification” includes a systematic study concerning the preparation of a bimetallic Pt-Fe catalyst (0.75% Pt) supported on a defective graphene surface, characterised by a high dispersion of Pt clusters in interaction with atomically-dispersed Fe atoms, featuring a superior CO PROX performance in the range of 30-200°C. In particular, the manuscript includes a paragraph devoted to the structural characterizations of PtFe cluster catalyst, documenting by HAADF-STEM, XANES and EXAFS techniques the construction of fully-exposed bimetallic Pt-Fe cluster catalysts through the deposition-precipitation method. Owing to the promoting effect of Fe atoms on the O-activation step, the Pt-Fe clusters achieve a complete elimination of CO through PROX reaction with almost 100% CO selectivity, while maintaining a good stability and almost unchanged physico-chemical characteristics. Theoretical calculations reveal that maximized atomic exposure and abundant interfacial sites facilitate oxygen activation on unsaturated Fe species and CO adsorption on electron-rich Pt clusters, favouring the CO PROX efficiency of the studied catalyst. Overall, the manuscript reports a good work, carried out with a systematic and logic methodological approach and satisfactorily discussed in the context of literature data. Although scientifically sound, however, novelty and originality of the topic, from both experimental and theoretical points of view, are not sufficiently high to deserve publication in Nature, also considering the (minor) improvement in activity in comparison to previous literature data (see Fig. 2C and Tab. S5).

Reviewer #2 (Remarks to the Author):

This is an interesting paper in which the authors report the preparation and characterization of fully-exposed Pt-Fe clusters on a hybrid nanodiamond/graphene support with efficient CO PROX performance. It is a significant and innovative work for the synthesis challenge of atomically dispersed bimetallic catalyst, which achieve maximized atomic efficiency and active interfacial site. On the other hand, the authors provide the insightful discussion about the catalytic behavior of fully-exposed Pt-Fe cluster, particularly by in-situ experiments (XPS and CO-DRIFTS) and DFT calculations. However, some parts of the discussion over experiments results were not clear and could be confusing. Thus, some revisions are needed before final publication.

Comments:

1. The novel bimetallic structure with atomically dispersed metal atoms were observed in Figure 1. The NDG support played an important role in preparing the PtFe catalyst. Why the NDG support could highly disperse the metal atoms? Is there a stronger interaction with metal? The author should give more discussion and evidence about NDG support.
2. The structure of “fully-exposed cluster” was mentioned. Please give more clear explanation of fully-exposed cluster. Whether it was based on metal-metal coordination number or specific cluster size range?
3. From the STEM images it clearly showed that the “atomically dispersed species” as well as “cluster

sized species” co-existed on the NDG support, which was also evidenced in the broad CO adsorption peak in DRIFTS characterization (Figure 4c). Could authors elucidate the mechanism of the formation of “bimetallic Pt-Fe cluster”?

4. The author mentioned in the paper the blue shift of linear CO indicated the co-adsorption of CO and O₂ on Pt or Pt-Fe interface. Explanation should be given.

5. The author concluded that “Pt-Fe cluster with maximized atomic utilization and abundant interfacial sites could facilitate O₂ activated by neighboring unsaturated Fe site and readily react with adsorbed CO on the fully-exposed Pt cluster”. However, except the DFT calculations, I did not see any experiment result to support this conclusion. The author should discuss more from DRIFTS results or design additional experiments (such as rate order measurement, TPD or TPSR experiments?) to elucidate this conclusion.

6. More STEM images of 0.75Pt0.2Fe/ND@G in figure S1 should be illustrated to prove the dispersion of PtFe species.

Reviewer #3 (Remarks to the Author):

Preferential oxidation of CO in presence of excess hydrogen is a mature area of research and there are several reports in the literature using even very cheaper metal Cu at very low temperature. Authors should focus what is new in their work. What is active species of Pt during reaction? Excess H₂ is present in the feed so the reducible nature of Pt species shown be checked by temperature programmed reduction (TPR) using H₂ (H₂-TPR). Authors used stoichiometric amount of O₂ (CO:O₂=2:1) in the reaction (Fig 3A) but they should use more than stoichiometric ratio (at least 1:2) to check the O₂ selectivity, as there is possibility that H₂ can also be oxidized. In the realistic PEMFC condition O₂ concentration is more than 1%. Activity of the catalyst should be checked using realistic feed composition in presence of CO₂ and H₂O. What is the role of Fe here? What is the stability of the individual catalyst 0.75% Pt on ND and 0.25%Fe?

Response to reviewers

REVIEWER COMMENTS:

Reviewer #1 (Remarks to the Author):

The manuscript “Fully-exposed Pt-Fe cluster for efficient preferential oxidation of CO towards hydrogen purification” includes a systematic study concerning the preparation of a bimetallic Pt-Fe catalyst (0.75% Pt) supported on a defective graphene surface, characterised by a high dispersion of Pt clusters in interaction with atomically-dispersed Fe atoms, featuring a superior CO PROX performance in the range of 30-200°C. In particular, the manuscript includes a paragraph devoted to the structural characterizations of PtFe cluster catalyst, documenting by HAADF-STEM, XANES and EXAFS techniques the construction of fully-exposed bimetallic Pt-Fe cluster catalysts through the deposition-precipitation method. Owing to the promoting effect of Fe atoms on the O-activation step, the Pt-Fe clusters achieve a complete elimination of CO through PROX reaction with almost 100% CO selectivity, while maintaining a good stability and almost unchanged physico-chemical characteristics.

Theoretical calculations reveal that maximized atomic exposure and abundant interfacial sites facilitate oxygen activation on unsaturated Fe species and CO adsorption on electron-rich Pt clusters, favouring the CO PROX efficiency of the studied catalyst. Overall, the manuscript reports a good work, carried out with a systematic and logic methodological approach and satisfactorily discussed in the context of literature data. Although scientifically sound, however, novelty and originality of the topic, from both experimental and theoretical points of view, are not sufficiently high to deserve publication in Nature, also considering the (minor) improvement in activity in comparison to previous literature data (see Fig. 2C and Tab. S5).

Response: We thank the reviewer for the nice comment. Indeed, the preferential oxidation of CO (PROX) reaction has been studied in the past. For instance, Pt-Fe/SiO₂ catalyst with Pt-Fe nanoparticles reported by Bao et al. (Science 2010, 328, 1141-1144), Pt/Fe(OH)_x catalyst with Pt nanocrystals and Fe-OH layers reported by Zheng et al. (Science 2014, 344, 495-499), and xFe-Pt/SiO₂ catalyst with Pt nanoparticles reported by Lu et al. (Nature 2019, 565, 631-635). One thing in common is that the active Pt species in the above-mentioned studies are all metal nanoparticles over 2 nm in size, which results in low metal utilization. As most efficient catalysts of PROX reaction remain noble metals, it is critical to maximize the utilization efficiency of noble metal and reduce the cost.

In this work, we firstly reported a fully-exposed bimetallic Pt-Fe cluster catalysts with both atomically dispersed Pt and Fe species, which is an original structure in comparison with previous reported Pt nanoparticle-based catalysts for PROX reaction. Meanwhile, it is novelty in maximizing metal efficiency and the number of Pt-Fe interfacial sites for bimetallic PtFe catalyst, leading to the superior catalytic performance. Moreover, considering the PEFMC cold-start operation condition, it is critical and breakthrough in superior oxidation activity at low temperature, even room temperature, for our Pt-Fe catalyst. Specifically, except for exceeding the highest activity reported in the literature (Nature 2019, 565, 631-635), there has been an obvious improvement in activity with other reported catalysts. More importantly, using in-situ experiments and theoretical calculation, we established a novel and stable Pt-Fe structure model and firstly proposed a reaction mechanism based on high CO coverage in revealing the reason for its excellent performance. We believe that the current work is of important scientific significance in the design of efficient catalysts and is novelty in experimental and theoretical views to publish in Nature Communications. Special thanks to the reviewer for his/her insight comments.

Reviewer #2 (Remarks to the Author):

This is an interesting paper in which the authors report the preparation and characterization of fully-exposed Pt-Fe clusters on a hybrid nanodiamond/graphene support with efficient CO PROX performance. It is a significant and innovative work for the synthesis challenge of

atomically dispersed bimetallic catalyst, which achieve maximized atomic efficiency and active interfacial site. On the other hand, the authors provide the insightful discussion about the catalytic behavior of fully-exposed Pt-Fe cluster, particularly by in-situ experiments (XPS and CO-DRIFTS) and DFT calculations. However, some parts of the discussion over experiments results were not clear and could be confusing. Thus, some revisions are needed before final publication.

Comments:

1. The novel bimetallic structure with atomically dispersed metal atoms were observed in Figure 1. The NDG support played an important role in preparing the PtFe catalyst. Why the NDG support could highly disperse the metal atoms? Is there a stronger interaction with metal? The author should give more discussion and evidence about NDG support.

Response: We thank the reviewer for the suggestion. The metal species (Pt, Sn, Cu et al.) were highly dispersed on the ND@G due to the strong interaction between metal and ND@G support in our previous works (J. Am. Chem. Soc. 2018, 140, 13142–13146; Nat. Commun. 2019, 10, 4431; Nat. Commun. 2021, 12, 2664.). Our group reported a work of Pt NPs/ND@G, including the discussion of strong metal–support interaction (SMSI) for the Pt/ND@G catalyst by the H₂-TPR experiment (ACS Catal. 2017, 7, 3349–3355). In the work, the reduction temperature of Pt/ND@G is higher than that of Pt/Al₂O₃ (Figure R1), attributed to a stronger interaction between the Pt species and ND@G. So, the NDG support played an important role in preparing the highly dispersed PtFe catalyst.

Figure R1. (a) temperature-programmed reduction (TPR) for Pt/Al₂O₃, Pt/ND@G (ACS Catal. 2017, 7, 3349–3355)

2. The structure of “fully-exposed cluster” was mentioned. Please give more clear explanation of fully-exposed cluster. Whether it was based on metal-metal coordination number or specific cluster size range?

Response: We thank the reviewer for the comment. The detailed description of fully exposed clusters could be followed in our recent review article (ACS Cent. Sci. 7, 262–273 (2021)). The fully exposed cluster catalysts (FECCs) have an ultrasmall size, normally below 1 nm, and a small contact angle with the support, forming a layered structure, so that they provide maximized atom utilization and possess rich active sites and easily identified coordination structures.

3. From the STEM images it clearly showed that the “atomically dispersed species” as well as “cluster sized species” co-existed on the NDG support, which was also evidenced in the broad CO adsorption peak in DRIFTS characterization (Figure 4c). Could authors elucidate the mechanism of the formation of “bimetallic Pt-Fe cluster”?

Response: We thank the reviewer for this constructive suggestion. We are sorry for missing the detailed discussion about the DRIFTS results. Specifically, for CO-DRIFT of 0.75Pt0.2Fe/ND@G catalyst, the broad CO adsorption peak was assigned at 2090cm⁻¹ and 2051cm⁻¹, which corresponds to linear CO adsorption on atomically dispersed Pt and linear CO adsorption on Pt clusters, respectively, which has been found in previous study (ACS Catal. 2019, 9, 5752–5759); ACS Catal. 2022, 12, 9602–9610); Science 2015, 350, 189-192). Pt-Fe clusters were synthesized using deposition precipitation method, where a certain amount of Pt and Fe-containing ions were simultaneously added into the dispersion with ND@G and weakly alkalis (NaCOOH). The resulting metal oxides uniformly dispersed on the ND@G surface were reduced at high temperature, and the platinum and iron atoms migrate and combine with each other to form stable bimetallic Pt-Fe clusters on the carbon surface.

Figure R2. In-situ DRIFTS study of CO adsorption during He purging on 0.75Pt0.2Fe/ND@G catalyst.

4. The author mentioned in the paper the blue shift of linear CO indicated the co-adsorption of CO and O₂ on Pt or Pt-Fe interface. Explanation should be given.

Response: We appreciate the reviewer for this insightful comment. After introduction of O₂, the band of the adsorbed CO shifted to a higher frequency, which reflects the decreased back-donation of electrons from Pt to CO, suggesting that the co-adsorption of CO and O₂ on Pt sites or Pt-Fe sites had occurred. (ACS Catal. 2014, 4, 2113–2117); Angew. Chem. Int. Ed. 2012, 51, 2920–2924)

5. The author concluded that “Pt-Fe cluster with maximized atomic utilization and abundant interfacial sites could facilitate O₂ activated by neighboring unsaturated Fe site and readily react with adsorbed CO on the fully-exposed Pt cluster”. However, except the DFT calculations, I did not see any experiment result to support this conclusion. The author should discuss more from DRIFTS results or design additional experiments (such as rate order measurement, TPD or TPSR experiments?) to elucidate this conclusion.

Response: We thank the reviewer for this constructive suggestion. For CO oxidation and PROX reactions on supported Pt-group metal catalysts, the activation of O₂ is critical because the CO adsorption on these metals is so strong that O₂ cannot competitively adsorb and be activated at low temperatures. In our work, the performance has been significantly improved

after the introduction of iron. According to our design strategy, in the case of the overall morphology, size and dispersion of the PtFe/ND@G catalyst like those of the Pt/ND@G catalyst, the crucial factor contributing to the observed high activity is the promotion of O₂ activation by the special iron species. In order to prove this reaction mechanism, we first performed *in-situ* XPS experiments (Figure 4f) and found that the valence state of Fe changed and existed in the form of Fe²⁺ after the introduction of reaction gas, indicating that iron species activated oxygen to cause the valence state change during the reaction. In addition, we adsorbed CO and oxygen simultaneously on the PtFe/ND@G catalyst using *in-situ* CO-DRIFT (Figure 4c-d), and compared the results with those of Pt/ND@G catalyst (Figure S25). It was found that the band of adsorbed CO shifted significantly after addition of O₂ for the Pt/ND@G catalyst, attributed to the competitive adsorption of O₂ and CO on the same Pt sites. However, this shift phenomenon is very slight for the PtFe/ND@G catalyst. Due to the existence of the Pt-Fe interface sites, the O₂ adsorbed most probably on the Fe atom of the interface, which has a weak effect on the Pt sites. Therefore, according to *in-situ* XPS and CO-DRIFT results, CO oxidation may follow the noncompetitive Langmuir–Hinshelwood mechanism, that is, CO adsorbs and is activated on the Pt sites whereas O₂ adsorbs on the neighboring unsaturated Fe sites and the reaction then takes place at the Pt-Fe interfaces.

6. More STEM images of 0.75Pt0.2Fe/ND@G in figure S1 should be illustrated to prove the dispersion of PtFe species.

Response: We appreciate the reviewer for the suggestion. The STEM images of 0.75Pt0.2Fe/ND@G catalyst are provided in Figure R3. The PtFe species are fully exposed on ND@G.

Figure R3. HAADF-STEM images of the 0.75Pt0.2Fe/ND@G catalyst.

Reviewer #3 (Remarks to the Author):

Preferential oxidation of CO in presence of excess hydrogen is a mature area of research and there are several reports in the literature using even very cheaper metal Cu at very low temperature. Authors should focus what is new in their work. What is active species of Pt during reaction? Excess H₂ is present in the feed so the reducible nature of Pt species shown be checked by temperature programmed reduction (TPR) using H₂ (H₂-TPR). Authors used stoichiometric amount of O₂ (CO:O₂=2:1) in the reaction (Fig 3A) but they should use more than stoichiometric ratio (at least 1:2) to check the O₂ selectivity, as there is possibility that H₂ can also be oxidized. In the realistic PEMFC condition O₂ concentration is more than 1%. Activity of the catalyst should be checked using realistic feed composition in presence of CO₂ and H₂O. What is the role of Fe here? What is the stability of the individual catalyst 0.75% Pt on ND and 0.25% Fe?

Response: We thank the reviewer for the comment. Indeed, several works about Cu catalysts, especially CuCe catalysts, have been reported for PROX reaction due to the low cost. The main research contents are to optimize the catalytic activity, explore the active structure of Cu catalyst including Cu valence and species, and reveal the reaction mechanism. However, among the Cu work reported so far, there is still a big activity gap between the Cu catalysts

and Pt-group metal catalysts at low temperatures, especially below 80 °C to room temperature (ACS Catal. 2016, 6, 3520–3530; Appl. Catal. B Environ. 2007, 72, 149-156). Besides, for the main study of CuCe catalyst, Cu loading is high with complex structure, and the stability is very poor in the presence of H₂O and CO₂, so that it does not meet the requirements of PROX application. In our work, a novel Pt-based catalyst is prepared, with the advantages of low loading, high metal utilization and low cost, and its low temperature activity is the best reported so far in PROX. The main active species are fully exposed PtFe clusters with single interface sites where Pt species is mainly formed by few-atom ensembles. In our previous work, the reducible nature of Pt/ND@G catalyst has been studied by H₂-TPR experiment (Figure R4). In addition, we carried out pre-reduction treatment at 400 °C before testing the reaction, and combined with XPS data, it was proved that the Pt species of the catalyst was mainly metallic, so that the reduction of Pt is not affected in the presence of excess hydrogen.

According to the reviewer's suggestion, we tested the performance of PROX under CO:O₂=1:2 and the complete CO removal could still be achieved at low temperature (below 50 °C) while the selectivity remained above 50% (Figure R5). We have measured the activity and stability at 30 °C after introducing H₂O and CO₂ at the same reaction condition (Figure S12). Compared with PROX reaction gas, there is almost no loss of activity in the presence of H₂O and CO₂ with good stability. After the introduction of Fe, Fe atoms combine with Pt atoms to form stable fully exposed clusters with highly active Pt-Fe interface sites, in which Fe promotes oxygen adsorption and activation to increase the CO oxidation activity in a noncompetitive L-H mechanism.

We added the stability of 0.75Pt/ND@G and 0.25Fe/ND@G in the PROX reaction at 30 °C (Figure R6). The 0.75Pt/ND@G catalyst had certain loss of activity after a long time of reaction, while 0.25Fe/ND@G remained inactive for a long time because it did not have CO oxidation activity.

Figure R4. (a) temperature-programmed reduction (TPR) for Pt/Al₂O₃, Pt/ND@G (ACS Catal. 2017, 7, 3349–3355)

Figure R5. CO conversion and selectivity of the 0.75Pt0.2Fe/ND@G catalyst in PROX reaction. Reaction conditions: 1%CO, 2%O₂ and 48% H₂ balanced in He; the space velocity is 45000 mL g⁻¹ h⁻¹.

Figure R6. Stability of the 0.75Pt/ND@G (a) and 0.2Fe/ND@G (b) catalyst in the PROX reaction under the high space velocity at 30 °C (Reaction conditions: 1%CO, 0.5%O₂ and 48% H₂ balanced in He; GHSV=180000 mL g⁻¹ h⁻¹).

REVIEWER COMMENTS

Reviewer #2 (Remarks to the Author):

The author well answered most of the comments, and the synthesis process as well as structural features of the catalysts were clarified. However, the discussion concerning the catalytic mechanism was still elusive and lack quantitative experiment design. For example:

1) The DRIFTS result in Figure 4c clearly indicated that several peaks existed upon immediate CO adsorption. How to interpret these adsorption peaks? During He purging the intensity of different peaks varied and did this phenomenon reveal CO-induced structural changes of adsorption sites? Or just reflect the adsorption strength of different species? In Figure 4c and Figure S25a, both of the broad peak contained different kinds of Pt-CO species (Pt1-CO/Ptn-CO...). Thus, it is not accurate to ascribe the broad peak to a single adsorption configuration and say that “the CO chemisorption peak located at 2061 cm⁻¹, assigned to linear CO on Pt clusters...” and made following comparisons.

2) Line 286 read “... resulting in noncompetitive L-H mechanism”. However, the author wrote “the shift of 2061 cm⁻¹ to 2067 cm⁻¹ indicate co-adsorption of CO and O₂ on Pt clusters”. Following the thread of thought, the shift of 2054 cm⁻¹ to 2051 cm⁻¹ should lead to the same conclusion.

Overall, the revised manuscript only briefly describes the role of Fe in the synthesized catalyst and the conclusions drawn from the DRIFTS experiment was not convincing enough. Additional thorough and quantitative experiments related to the kinetic behavior of the catalysts and the promotional role of Fe should be designed, and the discussion about results need to be more precise.

Reviewer #3 (Remarks to the Author):

The authors have addressed almost all the concerns of the reviewers in a reasonable manner so the revised manuscript is recommended for publication

Response to reviewers

REVIEWER COMMENTS:

Reviewer #2 (Remarks to the Author):

The author well answered most of the comments, and the synthesis process as well as structural features of the catalysts were clarified. However, the discussion concerning the catalytic mechanism was still elusive and lack quantitative experiment design. For example:

1) The DRIFTS result in Figure 4c clearly indicated that several peaks existed upon immediate CO adsorption. How to interpret these adsorption peaks? During He purging the intensity of different peaks varied and did this phenomenon reveal CO-induced structural changes of adsorption sites? Or just reflect the adsorption strength of different species? In Figure 4c and Figure S25a, both of the broad peak contained different kinds of Pt-CO species (Pt₁-CO/Pt_n-CO...). Thus, it is not accurate to ascribe the broad peak to a single adsorption configuration and say that “the CO chemisorption peak located at 2061 cm⁻¹, assigned to linear CO on Pt clusters...” and made following comparisons.

Response: We thank the reviewer for this constructive suggestion. We are sorry for the detailed discussion about the DRIFTS results. Typically, the DRIFT results in Figure 4c and Figure S25a were collected from 0 min to 25 min during He purging after CO adsorption. In this purging process, the CO adsorption peaks in 0 min existed main CO physical adsorption peaks and CO chemical adsorption peaks. And after 25 min, the intensity of CO adsorption peak was decreased to a broad peak only as CO chemical adsorption peak. Indeed, the broad peak contained Pt single atoms and clusters. Because the CO adsorption on Pt₁ was very weak and the corresponding peak was located at a high wavenumber (2090 cm⁻¹), the broad peak for 0.75Pt0.2Fe/ND@G catalyst consisted of a main peak at 2051 cm⁻¹ assigned as CO adsorbed on Pt clusters and a slight peak for CO adsorbed on Pt₁ (Figure R1). And the broad peak for 0.75Pt/ND@G catalyst consisted of a main peak at 2061 cm⁻¹ assigned as CO adsorbed on Pt clusters and a slight peak for CO adsorbed on Pt₁. We revised the manuscript about the DRIFT results discussion according to the suggestion of the reviewer.

Figure R1. In-situ DRIFTS study of CO adsorption during He purging on 0.75Pt0.2Fe/ND@G catalyst.

2) Line 286 read “... resulting in noncompetitive L-H mechanism”. However, the author wrote “the shift of 2061 cm^{-1} to 2067 cm^{-1} indicate co-adsorption of CO and O₂ on Pt clusters”. Following the thread of thought, the shift of 2054 cm^{-1} to 2051 cm^{-1} should lead to the same conclusion.

Response: We appreciate the reviewer for the suggestion. We are sorry to give the ambiguous conclusion about the DRIFT results. In fact, the slight shift of 2051 cm^{-1} to 2054 cm^{-1} for the PtFe catalyst is more likely due to the slight oxidation of Pt sites by oxygen adsorption on the Pt-Fe interface. According to the calculation of Pt₄Fe model for CO oxidation process (Figure R2), the stable Pt₄FeO including the Pt-O-Fe interface was formed as the actual active site, indicating that in the process of CO oxidation, the Pt-Fe interface of PtFe catalyst was transformed to the stable Pt-O-Fe specie by O₂ activated on Fe site. Therefore, the shift of CO adsorbed on Pt for PtFe catalyst was discovered. Besides, the special oxidation phenomenon of Pt species for PtFe catalyst during the CO oxidation reaction was observed in in-situ XPS of Pt (Figure 4e), explaining the shift of 2051 cm^{-1} to 2054 cm^{-1} in CO DRIFT as powerful evidence. Overall, the preferential oxygen adsorption on Fe site and the transform from the Pt-Fe to Pt-O-Fe interface during CO oxidation process lead to oxidized Pt sites,

further caused the shift of CO adsorbed on Pt sites, not due to the co-adsorption of CO and O₂. Meanwhile, due to the presence of Pt-Fe interface for 0.75Pt0.2Fe/ND@G, combined with CO-DRIFT, in-situ XPS and DFT calculation, the activated oxygen species on Fe atoms readily reacted with CO adsorbed on Pt clusters, compared with 0.75Pt/ND@G.

Figure R2. DFT calculated pathways of CO oxidation for the Pt₄Fe₁@Gr model.

Overall, the revised manuscript only briefly describes the role of Fe in the synthesized catalyst and the conclusions drawn from the DRIFTS experiment was not convincing enough. Additional thorough and quantitative experiments related to the kinetic behavior of the catalysts and the promotional role of Fe should be designed, and the discussion about results need to be more precise.

Response: We appreciate the reviewer for this insightful comment. For the role of Fe site in our Pt-Fe catalysts, we firstly discussed the relationship between the reactivity and Pt-Fe structures in different Fe contents using the precise structure analysis, including the activity and apparent activation energy. As for the catalysis behavior and promotional role of Fe, in fact, it has been study in many reported literatures about facilitated oxygen adsorption and dissociation on Fe sites, particularly including iron hydroxide in Zheng's group (Science 2014, 344, 495-499) and Fe oxides in Huber's group (Journal of Catalysis 2018, 358, 19-26). However, the critical challenge of PtFe catalysts is the change of fine active structures during the actual reaction process to understand the catalytic behavior of actual active sites. A representative work is atomically dispersed iron hydroxide anchored on the Pt nanoparticles revealed by in situ X-ray absorption fine-structure measurements (Nature 2019, 565, 631-635). In our work, combined with in-situ CO DRIFT and XPS, the DFT calculation revealed innovatively that the fully exposed bimetallic PtFe cluster existed low oxidation state of Pt and Fe during the CO oxidation reaction and could readily with CO and facilitate oxygen activation. We believe that the role of Fe is clearly revealed in the current work.

Special thanks to the reviewer for the suggestion, which really help us a lot in improving the manuscript.

Reviewer #3 (Remarks to the Author):

The authors have addressed almost all the concerns of the reviewers in a reasonable manner so the revised manuscript is recommended for publication

Response: We thank the reviewer for the nice comment. The suggestions really help us a lot to improve the manuscript.

REVIEWERS' COMMENTS

Reviewer #2 (Remarks to the Author):

The authors have answered all the concerns of the reviewer with reasonable detail, and modifications have been made in the revised manuscript. So this revised manuscript is recommended for publication.

Reviewer #2 (Remarks to the Author):

The authors have answered all the concerns of the reviewer with reasonable detail, and modifications have been made in the revised manuscript. So this revised manuscript is recommended for publication.

Response: We appreciate the reviewer for the nice comments, which really help us a lot in improving the manuscript.